# New genetic tools for mushroom body output neurons in *Drosophila*

**Gerald M Rubin\*, Yoshinori Aso\***

Janelia Research Campus, Howard Hughes Medical Institute, Ashburn, United States

**Abstract** How memories of past events influence behavior is a key question in neuroscience. The major associative learning center in *Drosophila*, the mushroom body (MB), communicates to the rest of the brain through mushroom body output neurons (MBONs). While 21 MBON cell types have their dendrites confined to small compartments of the MB lobes, analysis of EM connectomes revealed the presence of an additional 14 MBON cell types that are atypical in having dendritic input both within the MB lobes and in adjacent brain regions. Genetic reagents for manipulating atypical MBONs and experimental data on their functions have been lacking. In this report we describe new cell-type-specific GAL4 drivers for many MBONs, including the majority of atypical MBONs that extend the collection of MBON driver lines we have previously generated (Aso et al., 2014a; Aso et al., 2016; Aso et al., 2019). Using these genetic reagents, we conducted optogenetic activation screening to examine their ability to drive behaviors and learning. These reagents provide important new tools for the study of complex behaviors in *Drosophila*.

**\*For correspondence:**
rubing@janelia.hhmi.org (GMR);
asoy@janelia.hhmi.org (YA)

**Competing interest:** The authors declare that no competing interests exist.

## eLife assessment

This work advances on two Aso et al 2014 eLife papers to describe further resources that are **valuable** for the field. This paper identified and contributes additional MBON split-Gal4s, **convincingly** describing their anatomy, connectivity and function.

## Introduction

The mushroom body (MB) is the major site of associative learning in insects (reviewed in *Heisenberg, 2003*; *Modi et al., 2020*). In the MB of each *Drosophila* brain hemisphere, multiple modalities of sensory stimuli are represented by the sparse activity of ~2000 Kenyon cells (KCs) whose parallel axonal fibers form the MB lobes. The lobes are further divided into compartments by the innervation patterns of dopaminergic neuron (DAN) axons and mushroom body output neuron (MBON) dendrites. MBONs provide the convergence element of the MB's three-layer divergent-convergent circuit architecture and the outputs of the MBONs drive learned behaviors.

Whereas the dendrites of typical MBONs are strictly confined to the MB lobes, analysis of the *Drosophila* connectome in larvae (*Eichler et al., 2017*) and adults (*Li et al., 2020*; *Scheffer et al., 2020*) revealed a new class of 'atypical' MBONs, consisting of 14 cell types in adults, that have part of their dendritic arbors outside the MB lobes, allowing them to integrate input from KCs with other information (*Li et al., 2020*). Some atypical MBONs receive dendritic input from other MBONs. Several provide output onto DANs that innervate the MB to participate in recurrent networks. At least five make strong, direct synaptic contact onto descending neurons that drive motor action. Three provide strong direct connections to tangential neurons of the fan-shaped body of the central complex (CX). However, analysis of the behaviors mediated by atypical MBONs has been limited by the lack of genetic drivers needed to manipulate their activity.

Here, we report the generation and characterization of cell-type-specific split-GAL4 driver lines for the majority of the atypical MBONs. We also provide driver lines for two typical MBON types

for which cell-type-specific split-GAL4 drivers were not previously available, and improved drivers for several other MBONs. We demonstrate the use of these new split-GAL4 lines in two behavioral assays. Using a four-armed olfactory arena equipped with optogenetic LED panels (*Aso and Rubin, 2016*; *Pettersson, 1970*), we assessed the ability of the labeled neurons to act as the unconditioned stimulus in an olfactory learning assay, an indication of their regulation of the activity of DANs. We also measured the effects of their optogenetic activation on kinematic parameters relevant for olfactory navigation. These reagents provide important new tools for the study of complex behaviors in *Drosophila*.

## Results and discussion

### Generation and characterization of split-GAL4 lines for MBONs

We generated split-GAL4 genetic driver lines corresponding to MBON cell type using well-established methods (*Dionne et al., 2018*; *Luan et al., 2006*; *Pfeiffer et al., 2010*; *Tirian and Dickson, 2017*). The morphologies of the MBONs, produced by electron microscopic reconstruction, were used to search databases of light microscopic images to identify enhancers whose expression patterns might yield clean driver lines for that MBON when intersected (*Otsuna et al., 2018*; *Meissner et al., 2023*). We took advantage of an expanded set of starting reagents that were not available when *Aso et al., 2014a*, generated the original set of split-GAL4 drivers for the MB cell types; in addition to the ~7000 GAL4 expression patterns described in *Jenett et al., 2012*, we had access to an additional ~9000 GAL4 expression patterns (*Tirian and Dickson, 2017*). A total of approximately 600 intersections were experimentally tested to generate the split-GAL4 lines reported here.

*Figure 1* shows examples of expression patterns of some of the highest quality split-GAL4 lines. For many cell types we were able to identify multiple different split-GAL4 lines. The brain and ventral nerve cord expression patterns of all split-GAL4 lines are shown in *Figure 1—figure supplement 1* for atypical MBONs and *Figure 1—figure supplement 2* for typical MBONs. The original confocal stacks from which these figures were generated, as well as additional image data, are available for download at https://splitgal4.janelia.org. *Videos 1 and 2* provide examples of comparisons between light microscopic images from these lines and neuronal skeletons from the hemibrain dataset (*Scheffer et al., 2020*) that were used to confirm the assignment of cell-type identity. *Figure 1—figure supplement 3* summarizes what we consider to be the best available split-GAL4 lines for each of the MBON types identified by connectomics, based on a combination of the lines presented here and in previous studies. The expression patterns shown in this paper were obtained using an antibody against GFP which visualizes expression from 20xUAS-CsChrimson-mVenus in attP18. Directly visualizing the optogenetic effector is important since expression intensity, the number of labeled MBONs, and off-targeted expression can differ when other UAS-reporter/effectors are used (e.g. see Figure 2—figure supplement 1 of *Aso et al., 2014a*).

For typical MBONs, we provide split-GAL4 lines for two cell types for which drivers were not described in *Aso et al., 2014a*, MBON21 and MBON23. We also provide improved lines for MBON04 and MBON19; previous drivers for MBON04 also had expression in other MBON cell types and our new line for MBON19 has less off-target expression (see *Aso et al., 2014a*, for previous lines).

For atypical MBONs we were able to generate multiple, independent driver lines for MBON20, MBON29, MBON30, and MBON33 that provide strong and specific expression. Several lines for MBON31 were generated, but they are also stochastically expressed in MBON32. Lines for MBON26 and MBON35 have some off-target expression that might compromise their use in behavioral assays, but they should permit cell-type-specific imaging studies. We failed to generate lines for MBON24, MBON25, MBON27, MBON32, and MBON34. We identified two candidate lines for MBON28 but because the morphology of MBON28 is very similar to those of MBON16 and MBON17 we were not able to make a definitive cell-type assignment (see *Figure 1—figure supplement 3*).

### Activation phenotypes of MBON lines

MBONs are the first layer of neurons that transform memories stored inside the MB into actions. MBONs can also provide input to the dendrites of the DANs that innervate the MB lobes, forming a recurrent network. To investigate these two aspects of MBON function, we used a four-armed olfactory arena equipped with LED panels for optogenetics (*Figure 2A*). A group of ~20 starved flies that

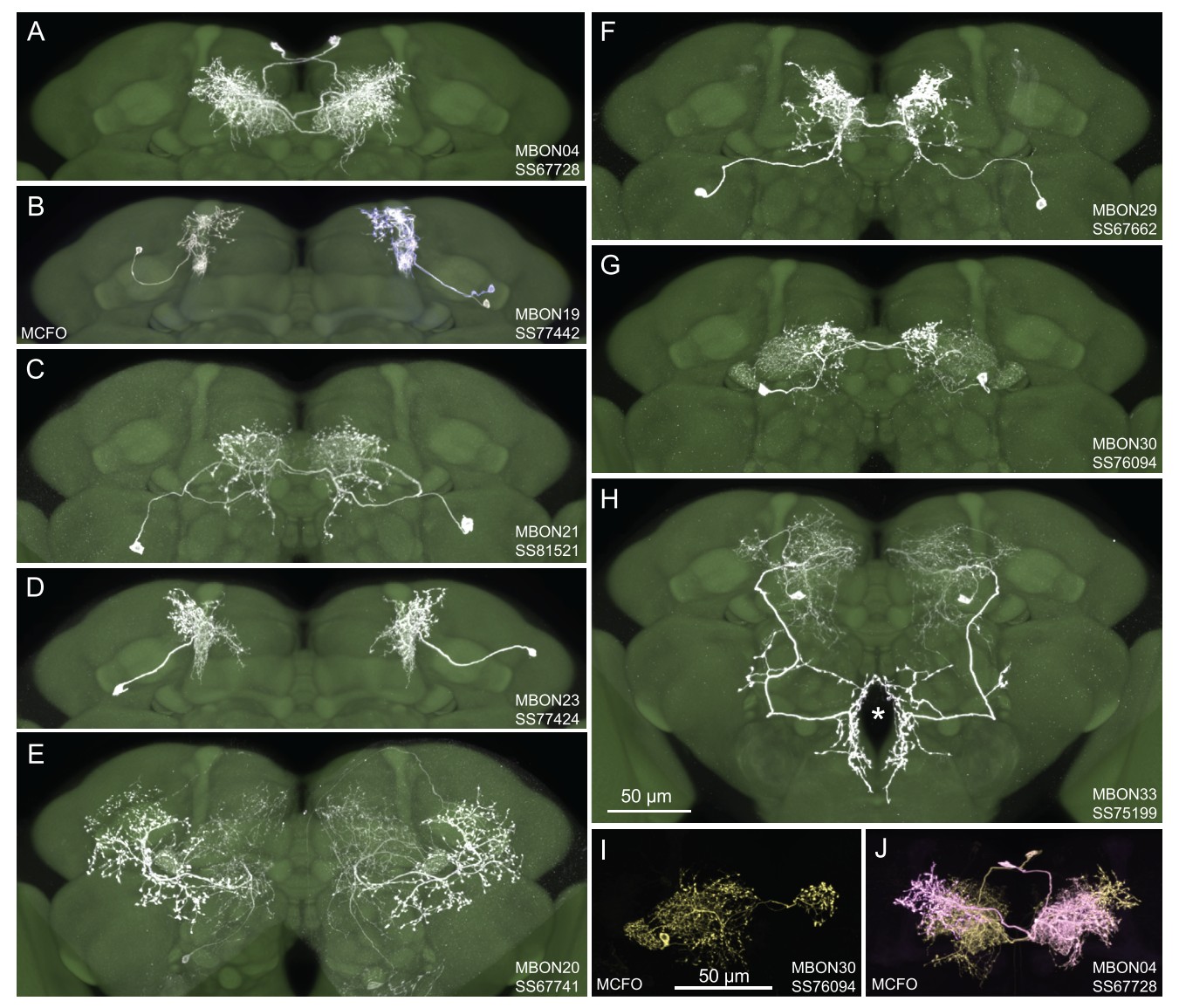

**Figure 1.** Selected images of new split-GAL4 lines. Panels A and C–H show expression (maximum intensity projections) of the indicated split-GAL4 line in the relevant portion of the brain. In panels A–H, the consensus JFRC2018 unisex brain template is also shown (green). Images showing the full brain, optic lobe, and ventral nerve cord of these lines can be found in *Figure 1—figure supplement 1* (for E–H) and *Figure 1—figure supplement 2* (for A–D). Panels B, I, and J show images derived from stochastic labeling that reveal the morphology of individual cells. The original confocal stacks from which these images were derived are available for download at https://splitgal4.janelia.org/cgi-bin/splitgal4.cgi.

The online version of this article includes the following figure supplement(s) for figure 1:

**Figure supplement 1.** Maximum intensity projections of the brains and ventral nerve cords of split-GAL4 lines for atypical mushroom body output neuron (MBON).

**Figure supplement 2.** Maximum intensity projections of the brains and ventral nerve cords of split-GAL4 lines for typical mushroom body output neuron (MBON).

**Figure supplement 3.** Summary list of selected split-GAL4 lines for all mushroom body output neuron (MBON) cell types.

express CsChrimson in a particular MBON cell type was subjected to a series of optogenetic olfactory conditioning and memory tests, and then to six trials of activation in the absence of odors but with airflow (*Figure 2B*). Using the same setup and similar protocols, we have previously characterized the dynamics of memories formed by optogenetically activating DAN cell types innervating different MB compartments (*Aso and Rubin, 2016*) and analyzed circuits downstream of the MBONs that promote

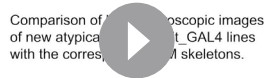

**Video 1.** Comparison of light microscopic images of atypical mushroom body output neurons (MBONs) with hemibrain skeletons of the corresponding cell types. https://elifesciences.org/articles/90523/figures#video1

upwind locomotion (*Aso et al., 2023*). *Figure 2C* displays the results of these behavioral experiments sorted by the mean memory score of each cell type in the final memory test. When possible, we ran multiple independent split-GAL4 lines for the same MBON cell type. The concurrence of phenotypes in these cases provides further evidence that the observed effects are due to the activation of that cell type rather than off-target expression or genetic background effects.

Training flies by paring odor presentation with activation of MBON21, using either of the two lines tested (SS81853 and SS81521), resulted in robust aversive memory (*Figure 2C*) as previously observed with another driver for this cell type that displayed weaker expression (see SS46348 in *Figure 1—figure supplement 2*; *Otto et al., 2020*). Both driver lines for the atypical MBON29 similarly induced aversive memory. These MBONs are both cholinergic, have dendrites in the γ4 and γ5 compartments, and synapse onto the dendrites of DANs that respond to punitive stimuli such as PAM-γ3 and PPL1-γ1pedc (*Figure 2C*, *Figure 2—figure supplement 1*; *Li et al., 2020*; *Otto et al., 2020*). In contrast, training flies with activation of MBON33 induced appetitive memory. MBON33 is also cholinergic but preferentially connects with octopaminergic neurons and reward-representing DANs. We noticed that confocal microscopy images of MBON33 visualized by split-GAL4 contain additional branches around the esophagus (*Figure 1H*), an area which was outside of EM hemibrain volume. Since octopaminergic neurons arborize in this area, the connection between MBON33 and octopaminergic neurons might be more extensive than the previously described using the hemibrain data (*Busch et al., 2009*; *Li et al., 2020*).

To explore kinematic parameters controlled by MBONs, we tracked the trajectories of individual flies during and after a 10 s optogenetic stimulus (*Figure 2C*). Although we used the same flies that went through the optogenetic olfactory conditioning, the activation phenotypes of positive control lines could be observed and was not compromised by their prior activation (*Figure 2—figure supplement 2*). In these assays we observed some variability between lines for the same cell type, presumably due to difference in expression level or off-targeted expression. Nevertheless, the two lines for MBON21 showed similar patterns of kinematic variables: a low walking speed in the presence of the red activation light, a stimulus that caused elevated locomotion in genetic control flies, and then orientation toward upwind when the optogenetic stimulus concluded (*Figures 2C and 3A–D*). Similar phenotypes were observed with a driver for a combination of the three glutamatergic MBON01, MBON03, and MBON04 (MB011B; *Figure 2C*). Despite their common anatomical features and memory phenotypes, MBON21 and MBON29 modulated distinct motor parameters. Neither of the two lines for MBON29 changed walking speed or orientation toward upwind when activated, but they both increased angular motion at the onset of activation, similar to the three lines for MBON26 (*Figure 3E–G*).

Finally, we asked if the MBON21, MBON29, and MBON33 lines that were able to serve as the unconditioned stimulus in memory formation also drove corresponding avoidance or attraction. Previous studies and the results shown in *Figure 2C* indicated that these are not always shared phenotypes; for example, the set of glutamatergic MBONs in MB011B whose dendrites lie in the γ5 and β'2 compartments can drive

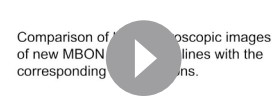

**Video 2.** Comparison of light microscopic images of typical mushroom body output neurons (MBONs) with hemibrain skeletons of the corresponding cell types. https://elifesciences.org/articles/90523/figures#video2

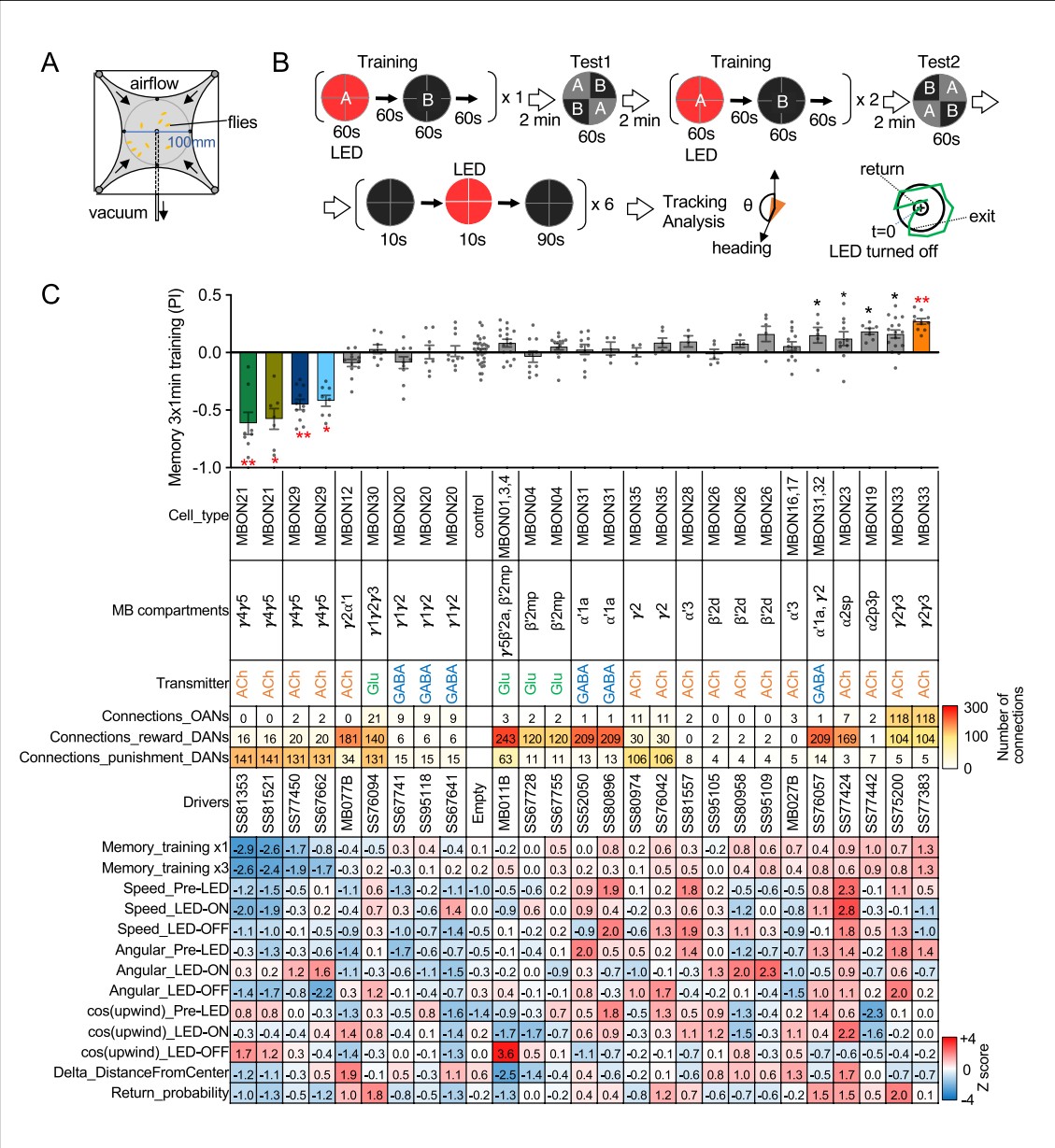

**Figure 2.** Behavioral consequences of optogenetic activation. (**A**) The four-armed olfactory arena. Approximately 20 starved female flies were confined in 10 cm diameter and 3 mm high circular area with a hole at the center for air suction. Odor was introduced through four channels at the corners. (**B**) The protocol for behavioral experiments. Flies were trained by pairing 60 s of odor A with 30 1 s pulses of 627 nm LED light, each separated by 1 s without illumination. A different odor, odor B, was presented without red LED illumination, and then preference between two odors was tested. In the reciprocal experiments, odor B was paired with red light and A was unpaired. The same training was repeated twice more and then a second odor preference test was performed. Finally, six cycles of 10 s 627 nm illumination were applied, spaced by 100 s intervals without odor. Airflow was maintained at 400 mL/min throughout the experiment. (**C**) Top: The memory scores at the second odor preference test, measured as preference indexes: [(number of flies in the paired odor quadrants)-(number of flies in the unpaired odor quadrants)]/total number of flies during the last 30 s of the 60 s test period. The red asterisks * and ** indicate p<0.05 or p<0.01, respectively: Dunn's multiple comparison tests compared to empty-split-GAL4 control, following Kruskal-Wallis test. The black * indicates p<0.05 without correction for multiple comparison. N=34 for the empty-split-GAL4 line and N=4–16 for other lines. All the lines were initially tested for four reciprocal experiments; lines with mean preference index above 0.1 or below –0.1 were subjected to additional tests. Cell types, the mushroom body (MB) compartments in which their dendrites lie, their neurotransmitters, the number of synaptic connections they make with dopaminergic (DANs) and octopaminergic (OANs) neurons, and the split-GAL4 driver lines used for the behavioral assays are designated. A summary of connections from all mushroom body output neuron (MBON) subtypes to DANs thought to signal reward or punishment and to OANs is shown in *Figure 2—figure supplement 1A*. Bottom: Z-scores [(values-mean)/standard deviation] for each parameter: speed, walking speed; angular, absolute of angular change relative to the previous frame at 30 FPS; cos(upwind), cosine of the fly's orientation toward the upwind direction (i.e. facing away from the center of the arena). ON periods correspond to the first 2 s of the 10 s LED ON periods, whereas OFF

*Figure 2 continued on next page*

*Figure 2 continued*

periods are the 2 s immediately after the LEDs were turned off. Delta_DistanceFromCenter is change in fly's mean distance from the center of the arena relative to its position at the onset of LED illumination. Return is a measure of the probability that flies return to the position that they occupied at the end of the LED stimulus. Flies are judged to have returned if they move 10 mm or more from their original position and then return to within 3 mm of the original position within 15 s.

The online version of this article includes the following source data and figure supplement(s) for figure 2:

**Source data 1.** The values used for *Figure 2*.

**Figure supplement 1.** Direct connections from mushroom body output neurons (MBONs) to dopaminergic neurons (DANs) and octopaminergic neurons (OANs).

**Figure supplement 2.** Activation phenotypes are not compromised by prior optogenetic olfactory conditioning.

downwind locomotion and avoidance behaviors (*Aso et al., 2023*; *Aso et al., 2014b*; *Matheson et al., 2022*) but do not induce aversive memory. We tested if flies expressing CsChrimson in each of these three MBON cell types prefer quadrants of the arena with red activating light during the first and second 30 s test periods (*Figure 3H1*). When CsChrimson is expressed in MBON21 or MBON29, flies avoid illuminated quadrants of the arena. Conversely, CsChrimson activation in a line for MBON33 promoted attraction to the illuminated quadrants, although this effect was observed only at the first test period. Thus, in the case of these three MBON cell types, memory formation and avoidance/attraction behaviors are correlated.

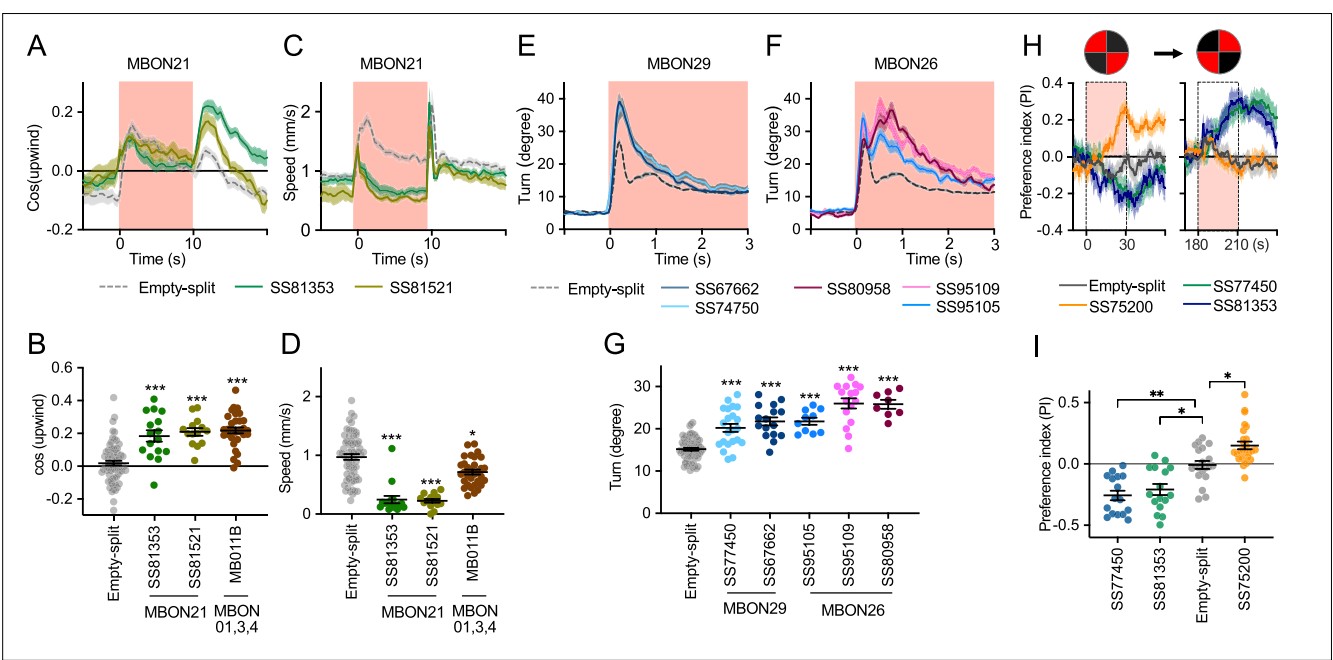

**Figure 3.** Additional behavioral consequences of optogenetic activation. (**A**) Time course of mean cos(upwind angle) for flies that express CsChrimson in MBON21 with designated drivers. The trace of empty-split-GAL4 is also shown. All the trajectories from six trials of movies were pooled to calculate a mean for each group of flies. Lines and shadings are means and SEMs. (**B**) Mean cos(upwind angle) during 2 s periods immediately after LED was turned off. (**C–D**) Time course and mean walking speed during 10 s LED ON period. (**E–F**) The mean cumulative turning angles in five movie frames of (total elapsed time 150 ms) for flies expressing CsChrimson in MBON29 and MBON26. (**G**) The cumulative turning angle during the first 2 s of LED ON period. (**A–G**) show data from the experiments described in *Figure 2*. (**H**) Preference for quadrants with red light. Flies expressing CsChrimson in MBON21, MBON29, or MBON33 were tested with 30 s continuous light of 627 nm LED in two quadrants. The test was performed a second time with illumination in opposite quadrants after a 150 s recovery period. (**I**) Mean preference index to the quadrants with red light during the last 5 s of two 30 s test periods. Dunn's multiple comparison tests compared to empty-split-GAL4 control, following Kruskal-Wallis test. *, **, and *** indicate p<0.05, p<0.01, or p<0.001, respectively: N=66 for Empty-split-GAL4 and 8–22 for other lines in (**A–G**). N=16–26 in (**H–I**).

The online version of this article includes the following source data for figure 3:

**Source data 1.** The values used for *Figure 3*.

## Concluding remarks

We generated and anatomically characterized an improved set of genetic driver lines for MBONs and provide the first driver lines for atypical MBONs. We expect these lines to be useful in a wide range of studies of fly behavior. We demonstrate the suitability of these lines for behavioral analyses by showing that multiple independent lines for the same cell type gave consistent results when assayed for their ability to serve as the unconditioned stimulus in memory formation and to produce appetitive or aversive movements. MBON21, MBON29, and MBON33, characterized in this study, have distinct features compared to well-studied MBONs that innervate the same compartments. MBON21 and MBON29 form cholinergic connections to the dendrites of DANs known to respond to punishment, whereas other MBONs from the same compartments form glutamatergic connections with reward-representing DANs (*Figure 4A*).

While most of sensory input to the MB is olfactory, the connectome revealed that two specific classes of KCs receive predominantly visual input. MBON19 provides the major output from one of these classes, KC α/βp, and about half of MBON19's sensory input is estimated to be visual. MBON33 provides a major output from the other class of visual KCs, KC γd, with more than half of the sensory input to its dendrites in the MB estimated to be visual. The cell-type-specific driver lines we provide for MBON19 and MBON33 should facilitate studies of the behavioral roles of the two streams of visual information that pass through the MB.

The MB and the CX are thought to play key roles in the most cognitive processes that the fly can perform including learning, memory, and spatial navigation. Two of the MBONs for which we generated cell-type-specific driver lines, MBON21 and MBON30, provide the strongest direct inputs to the CX from the MB (*Figure 4B*), while MBON30 receives over 3% of its input (450 synapses) from the CX cell type FR1 (aka FB2-5RUB). The genetic reagents we report here should advance studies of reinforcement signals in parallel memory systems, the role of visual inputs to the MB, and information flow from the MB to the CX.

# Materials and methods

## Flies

Split-GAL4 lines were created as previously described (*Dionne et al., 2018*). Flies were reared on standard cornmeal molasses food at 21–22°C and 50% humidity. For optogenetic activation experiments, flies were reared in the dark on standard food supplemented with retinal (Sigma-Aldrich, St. Louis, MO, USA) unless otherwise specified, 0.2 mM all trans-retinal prior to eclosion and 0.4 mM all trans-retinal post eclosion. Female flies were sorted on cold plates and kept in retinal food vials for at least 1 day prior to be transferred to agar vials for 48–72 hr of starvation. Flies were 4–10 days of age at the time of behavioral experiments. Images of the original confocal stacks for each of the lines are available at https://splitgal4.janelia.org.

The correspondence between the morphologies of EM skeletons and light microscopic images of GAL4 driver line expression patterns was used to assign GAL4 lines to particular cell types. This can be done with confidence when there are not multiple cell types with very similar morphology. However, in the case of MBON28 we were not able to make a definitive assignment because of the similarity in the morphologies of MBON16, MBON17, and MBON28.

## Immunohistochemistry and imaging

Dissection and immunohistochemistry of fly brains were carried out as previously described (*Aso et al., 2014a*). Each split-GAL4 line was crossed to the same Chrimson effector used for behavioral analysis. Full step-by-step protocols can be found at https://www.janelia.org/project-team/flylight/protocols. For single-cell labeling of neurons from selected split-Gal4 lines, we used the MultiColor FlpOut (MCFO) technique (*Nern et al., 2015*). *Video 1* and *Video 2* were produced using VVD Viewer (https://github.com/JaneliaSciComp/VVDViewer; copy archived at *JaneliaSciComp, 2024*) to generate a video comparing light and EM data (hemibrain v1.2) for each cell type. Individual videos were then concatenated, and text added using Adobe Premiere Pro.

## Optogenetics and olfactory learning assays

Groups of approximately 20 female flies were trained and tested at 25°C at 50% relative humidity in the fully automated olfactory arena for optogenetics experiments as previously described (*Aso and Rubin,*

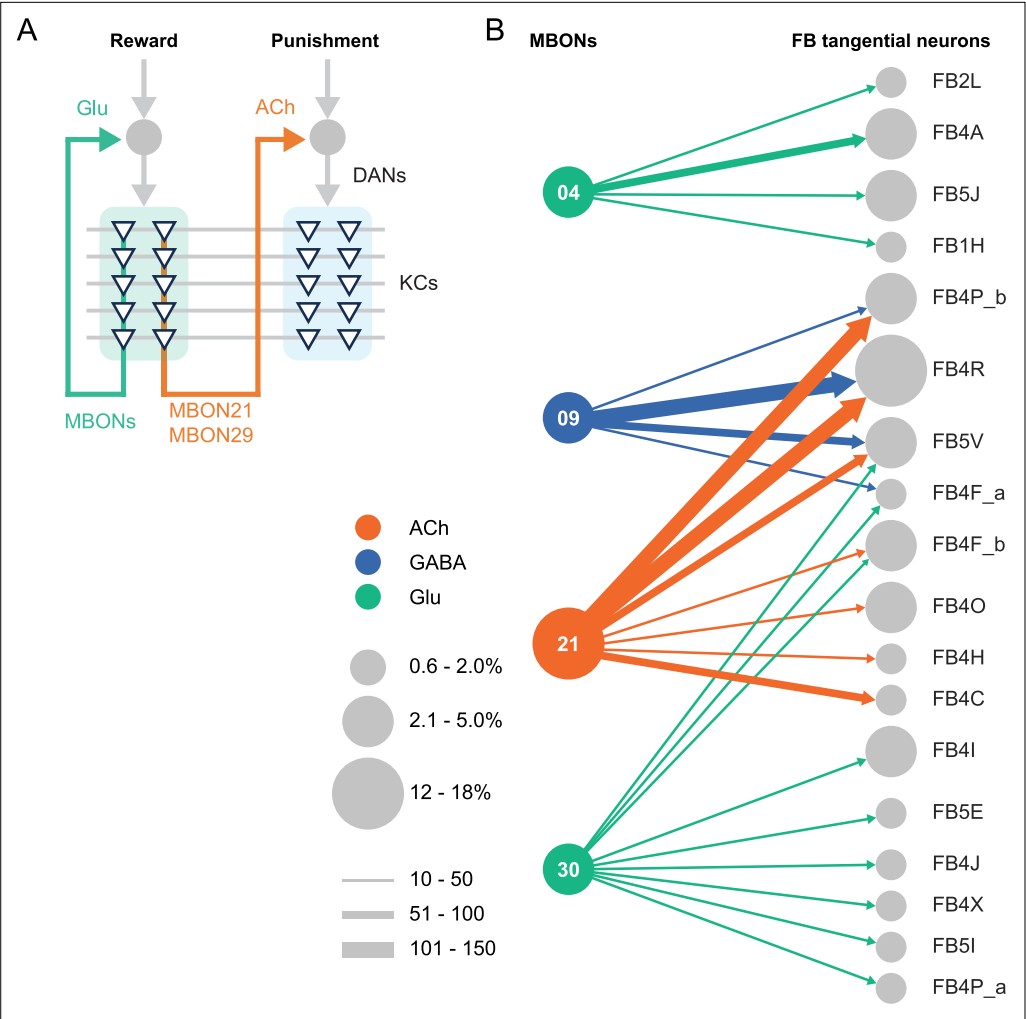

**Figure 4.** Diagrammatic summary of key outputs from selected mushroom body output neurons (MBONs). (**A**) MBON21 and MBON29 arborize dendrites in the g4 and g5 compartments that are innervated by reward representing dopaminergic neurons (DANs). MBON21 and MBON29 are cholinergic and preferentially connect with DANs that innervate other compartments to represent punishment, whereas other glutamatergic MBONs from these same compartments preferentially form connections with reward-representing DANs going back to the same compartments. In fly brains, acetylcholine (ACh) is largely excitatory via nicotinic ACh receptors, although the type A muscarinic ACh receptor can mediate an inhibitory effect (*Bielopolski et al., 2019*; *Manoim et al., 2022*). Glutamate (Glu) can be inhibitory or excitatory depending on the receptors in the postsynaptic cells. Glutamate is known to be inhibitory via the glutamate-gated chloride channel (GluClα) in the olfactory system (*Liu and Wilson, 2013*). All of the 10 types of DANs examined with RNA-seq express GluClα and Nmdar2 at high levels whereas expression of Nmdar1 and other glutamate receptors were limited and cell type specific (*Aso et al., 2019*). Results in some studies support an excitatory effect of at least a subset of glutamatergic MBONs on DANs (*Cohn et al., 2015*; *Ichinose et al., 2015*; *Otto et al., 2020*; *Zhao et al., 2018a*), while electrophysiological recordings identified inhibitory connection between glutamatergic MBON and the downstream interneurons (*Aso et al., 2023*). (**B**) Diagram showing direct connections between the mushroom body (MB) and central complex (CX) mediated by MBON21, MBON30, MBON09, and MBON04; these MBONs rank first, second, fifth, and seventh, respectively, based on the number of direct synaptic connections to the CX; numbers reflect connections between right hemisphere MBONs and right hemisphere FB tangential cells. For circles representing MBONs, the circle diameter represents the fraction of that MBONs direct output that goes to the CX. For the downstream neurons in the CX, circle diameter represents the fraction of that cell types direct input that comes from MBONs. Arrow width reflects synapse number. See Figure 19 of *Li et al., 2020* and Figure 46 of *Hulse et al., 2021* for additional information on the complete set of MB to CX connections.

*2016*; *Pettersson, 1970*; *Vet et al., 1983*). The 627 nm peak LED light was used at 22.7 µW/mm$^2$. The odors were diluted in paraffin oil (Sigma-Aldrich): pentyl acetate (1:10,000) and ethyl lactate (1:10,000). Videography was performed at 30 frames per second with 850 nm LED backlight with 820 nm longpass filter and analyzed using the Flytracker (*Wilson et al., 2013*) and Fiji (*Schindelin et al., 2012*).

## Statistics

Statistical comparisons were performed using the Kruskal-Wallis test followed by Dunn's post-test for multiple comparison (Prism; GraphPad Inc, La Jolla, CA, USA). Appropriate sample size for olfactory learning experiment was estimated based on the standard deviation of performance index in previous study using the same assay (*Aso and Rubin, 2016*).

## Connectomics

Information on connection strengths are taken from neuprint.janelia.org (hemibrain v1.2.1).

## Data availability

The confocal images of expression patterns are available online (http://www.janelia.org/split-gal4). The source data for figures are included in the manuscript.

## Acknowledgements

We thank the Janelia Fly Facility for help with fly husbandry and the FlyLight Project Team for dissection, histological preparation, and imaging of nervous systems. Marisa Dreher (Dreher Design Studios, Inc) assembled the videos and helped with figure design. Claire Managan segmented the neuron morphologies shown in the videos. Masayoshi Ito helped identify lines to use in intersections. Daisuke Hattori, Glenn Turner, Vivek Jayaraman, Toshihide Hige, and Yichun Shuai provided comments and suggestions on the early version of the manuscript.

## Additional information

### Funding

| Funder | Grant reference number | Author |
| --- | --- | --- |
| Howard Hughes Medical Institute | | Gerald M Rubin Yoshinori Aso |

The funders had no role in study design, data collection and interpretation, or the decision to submit the work for publication.

### Author contributions

Gerald M Rubin, Conceptualization, Resources, Data curation, Formal analysis, Supervision, Funding acquisition, Investigation, Visualization, Writing - original draft, Project administration, Writing - review and editing; Yoshinori Aso, Conceptualization, Formal analysis, Investigation, Visualization, Writing - original draft, Writing - review and editing

### Author ORCIDs

Gerald M Rubin (iD) http://orcid.org/0000-0001-8762-8703
Yoshinori Aso (iD) http://orcid.org/0000-0002-2939-1688

Reviewer #1 (Public Review): https://doi.org/10.7554/eLife.90523.3.sa1
Reviewer #2 (Public Review): https://doi.org/10.7554/eLife.90523.3.sa2
Author Response https://doi.org/10.7554/eLife.90523.3.sa3

## Additional files

### Supplementary files

• MDAR checklist

## Data availability

The confocal images of expression patterns are available online (http://www.janelia.org/split-gal4). Figure 2 - Source data 1 and Figure 3-source data 1 contains the numerical data used to generate figures.

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

# Appendix 1

**Appendix 1—key resources table**

| Reagent type (species) or resource | Designation | Source or reference | Identifiers | Additional information |
|---|---|---|---|---|
| Strain, strain background (*Drosophila melanogaster*) | Split-GAL4 lines | This paper, *Aso et al., 2014a*; PMID: 25535793 | https://splitgal4.janelia.org/cgi-bin/splitgal4.cgi | Available from Aso lab and Rubin lab |
| Strain, strain background (*Drosophila melanogaster*) | 20xUAS-CsChrimson- mVenus attP18 | *Klapoetke et al., 2014*; PMID: 24509633 | N.A. | |
| Strain, strain background (*Drosophila melanogaster*) | pJFRC200-10xUAS- IVS-myr::smGFP-HA in attP18 | *Nern et al., 2015*; PMID: 25964354 | N.A. | |
| Strain, strain background (*Drosophila melanogaster*) | pJFRC225-5xUAS- IVS-myr::smGFP-FLAG in VK00005 | *Nern et al., 2015*; PMID: 25964354 | N.A. | |
| Strain, strain background (*Drosophila melanogaster*) | pBPhsFlp2::PEST in attP3 | *Nern et al., 2015*; PMID: 25964354 | N.A. | |
| Strain, strain background (*Drosophila melanogaster*) | pJFRC201-10XUAS-FRT>STOP > FRT-myr::smGFP-HA in VK0005 | *Nern et al., 2015*; PMID: 25964354 | N.A. | |
| Strain, strain background (*Drosophila melanogaster*) | pJFRC240-10XUAS-FRT>STOP > FRT-myr::smGFP-V5-THS-10XUAS-FRT>STOP > FRT-myr::smGFP-FLAG_ in_su(Hw)attP1 | *Nern et al., 2015*; PMID: 25964354 | N.A. | |
| Strain, strain background (*Drosophila melanogaster*) | empty-split-GAL4 (p65ADZp attP40, ZpGAL4DBD attP2) | *Hampel et al., 2015*; PMID: 26344548 | RRID:BDSC_79603 | |
| Antibody | Anti-GFP (rabbit polyclonal) | Invitrogen | A11122 RRID:AB_221569 | 1:1000 |
| Antibody | Anti-Brp (mouse monoclonal) | Developmental Studies Hybridoma Bank | nc82 RRID:AB_2341866 | 1:30 |
| Antibody | Anti-HA-Tag (mouse monoclonal) | Cell Signaling Technology | C29F4; #3724 RRID:AB_10693385 | 1:300 |
| Antibody | Anti-FLAG (rat monoclonal) | Novus Biologicals | NBP1-06712 RRID:AB_1625981 | 1:200 |
| Antibody | Anti-V5-TAG Dylight-549 (mouse monoclonal) | Bio-Rad | MCA2894D549GA RRID:AB_10845946 | 1:500 |
| Antibody | Anti-mouse IgG(H&L) AlexaFluor-568 (goat polyclonal) | Invitrogen | A11031 RRID:AB_144696 | 1:400 |
| Antibody | Anti-rabbit IgG(H&L) AlexaFluor-488 (goat polyclonal) | Invitrogen | A11034 RRID:AB_2576217 | 1:800 |
| Antibody | Anti-mouse IgG(H&L) AlexaFluor-488 conjugated (donkey polyclonal) | Jackson Immuno Research Labs | 715-545-151 RRID:AB_2341099 | 1:400 |
| Antibody | Anti-rabbit IgG(H&L) AlexaFluor-594 (donkey polyclonal) | Jackson Immuno Research Labs | 711-585-152 RRID:AB_2340621 | 1:500 |
| Antibody | Anti-rat IgG(H&L) AlexaFluor-647 (donkey polyclonal) | Jackson Immuno Research Labs | 712-605-153 RRID:AB_2340694 | 1:300 |
| Antibody | Anti-rabbit IgG(H+L) Alexa Fluor 568 (goat polyclonal) | Invitrogen | A-11036 RRID:AB_10563566 | 1:1000 |
| Chemical compound, drug | Pentyl acetate | Sigma-Aldrich | 109584 | 1:10,000 in paraffin oil |
| Chemical compound, drug | Ethyl lactate | Sigma-Aldrich | W244015 | 1:10,000 in paraffin oil |
| Chemical compound, drug | Paraffin oil | Sigma-Aldrich | 18512 | |
| Software, algorithm | ImageJ and Fiji | NIH; *Schindelin et al., 2012* | https://imagej.nih.gov/ij/ http://fiji.sc/ | |

*Appendix 1 Continued on next page*

*Appendix 1 Continued*

| Reagent type (species) or resource | Designation | Source or reference | Identifiers | Additional information |
|---|---|---|---|---|
| Software, algorithm | MATLAB | MathWorks | https://www.mathworks.com/ | |
| Software, algorithm | Adobe Illustrator CC | Adobe Systems | https://www.adobe.com/products/illustrator.html | |
| Software, algorithm | GraphPad Prism 9 | GraphPad Software | https://www.graphpad.com/scientific-software/prism/ | |
| Software, algorithm | Caltech FlyTracker | *Eyjolfsdottir et al., 2014* | https://github.com/kristinbranson/FlyTracker | |
| Software, algorithm | neuPrint | *Plaza et al., 2022* | https://neuprint.janelia.org/ | |
| Software, algorithm | Cytoscape | *Shannon et al., 2003* | https://cytoscape.org/ | |
| Software, algorithm | Janelia workstation | HHMI Janelia | https://doi.org/10.25378/janelia.8182256.v1 | |
| Software, algorithm | NeuTu | *Zhao et al., 2018b*; *Zhao et al., 2018c* | https://github.com/janelia-flyem/NeuTu | |
| Software, algorithm | VVD Viewer | *Wan et al., 2012*; *Kawase et al., 2012* | https://github.com/takashi310/VVD_Viewer | |
| Other | Grade 3 MM Chr Blotting Paper | Whatmann | 3030-335 | Used in glass vials with paraffin oil diluted odors |
| Other | Mass flow controller | Alicat | MCW-200SCCM-D | Mass flow controller used for the olfactory arena |

