## [Editor Report · eLife assessment]

This work advances on two Aso et al 2014 eLife papers to describe further resources that are **valuable** for the field. This paper identified and contributes additional MBON split-Gal4s, **convincingly** describing their anatomy, connectivity and function.

---

## [Referee Report · Reviewer #1 (Public Review)]

In this manuscript Rubin and Aso provide important new tools for the study of learning and memory in *Drosophila*. In flies, olfactory learning and memory occurs at the Mushroom Body (MB) and is communicated to the rest of the brain through Mushroom Body Output Neurons (MBONs). Previously, typical MBONs were thoroughly studied. Here, atypical MBONs that have dendritic input both within the MB lobes and in adjacent brain regions are studied. The authors describe new cell-type-specific GAL4 drivers for the majority of atypical MBONs (and other MBONs) and using an optogenetic activation screen they examined their ability to drive behaviors and learning.

The experiments in this manuscript were carefully performed and the results are clear. The tools provided in this manuscript are of great importance to the field.

---

## [Referee Report · Reviewer #2 (Public Review)]

In this study, Aso and Rubin generated new split-GAL4 lines to label *Drosophila* mushroom body output neurons (MBONs) that previously lacked specific GAL4 drivers. The MBONs represent the output channels for the mushroom body (MB), a computational center in the fly brain. Prior research identified 21 types of typical MBONs whose dendrites exclusively innervate the MB and 14 types of atypical MBONs whose dendrites also innervate brain regions outside the MB. These MBONs transmit information from the MB to other brain areas and form recurrent connections to dopaminergic neurons whose axonal terminals innervate the MB. Investigating the functions of the MBONs is crucial to understanding how the MB processes information and regulates behavior. The authors previously established a collection of split-GAL4 lines for most of the typical MBONs and one atypical MBON. That split-GAL4 collection has been an invaluable tool for researchers studying the MB. This work extends their previous effort by generating additional driver lines labeling the MBON types not covered by the previous split-GAL4 collection. Using these new driver lines, the authors also activated the labeled MBONs using optogenetics and assessed their role in learning, locomotion, and valence coding. The expression patterns of the new split-GAL4 lines and the behavioral analysis presented in this manuscript are convincing. I believe that these new lines will be a valuable resource for the fly community.

---

## [Author Response]

The following is the authors’ response to the original reviews.

Thank you for submitting your article "New genetic tools for mushroom body output neurons in *Drosophila*" for consideration by eLife. Your article has been reviewed by 2 peer reviewers, and the assessment has been overseen by a Reviewing Editor and Albert Cardona as the Senior Editor.
**eLife assessment:**
This work advances on two Aso et al 2014 eLife papers to describe further resources valuable for the field. This paper adds more MBON split-Gal4s convincingly describing their anatomy, connectivity and function.
**Public Reviews:**

**Reviewer #1 (Public Review):**
In this manuscript Rubin and Aso provide important new tools for the study of learning and memory in *Drosophila*. In flies, olfactory learning and memory occurs at the Mushroom Body (MB) and is communicated to the rest of the brain through Mushroom Body Output Neurons (MBONs). Previously, typical MBONs were thoroughly studied. Here, atypical MBONs that have dendritic input both within the MB lobes and in adjacent brain regions are studied. The authors describe new cell-type-specific GAL4 drivers for the majority of atypical MBONs (and other MBONs) and using an optogenetic activation screen they examined their ability to drive behaviors and learning.The experiments in this manuscript were carefully performed and the results are clear. The tools provided in this manuscript are of great importance to the field.
**Reviewer #2 (Public Review):**
In this study, Aso and Rubin generated new split-GAL4 lines to label *Drosophila* mushroom body output neurons (MBONs) that previously lacked specific GAL4 drivers. The MBONs represent the output channels for the mushroom body (MB), a computational center in the fly brain. Prior research identified 21 types of typical MBONs whose dendrites exclusively innervate the MB and 14 types of atypical MBONs whose dendrites also innervate brain regions outside the MB. These MBONs transmit information from the MB to other brain areas and form recurrent connections to dopaminergic neurons whose axonal terminals innervate the MB. Investigating the functions of the MBONs is crucial to understanding how the MB processes information and regulates behavior. The authors previously established a collection of split-GAL4 lines for most of the typical MBONs and one atypical MBON. That split-GAL4 collection has been an invaluable tool for researchers studying the MB. This work extends their previous effort by generating additional driver lines labeling the MBON types not covered by the previous split-GAL4 collection. Using these new driver lines, the authors also activated the labeled MBONs using optogenetics and assessed their role in learning, locomotion, and valence coding. The expression patterns of the new split-GAL4 lines and the behavioral analysis presented in this manuscript are generally convincing. I believe that these new lines will be a valuable resource for the fly community.
**Recommendations for the authors:**
Minor additional suggestions:1. Please ensure that the FlyLight links are provided for the new splitGal4s in the methods as well as results.

We added the requested link to the methods.

2. Correct a typo in 'ethyl lactate in the learning assays section of methods

corrected

**Reviewer #1 (Recommendations For The Authors):**
In the behavior assay, the authors use the same flies that were used for optogenetic olfactory conditioning and memory tests, to also examine the effects of activation in the absence of odors but with airflow. I think this may affect the interpretation of the results. If possible, it would be nice to show in the MBON types where a conditioning effect was found (i.e. MBON21, 29, 33) that performing the activation in the absence of odors but with airflow without previous conditioning yields the same results.

We share the reviewers concern that behavioral phenotypes during the later 10s LED sessions may be compromised by early optogenetic olfactory conditioning. Therefore, prior to running the experiment shown in Figure 2, we confirmed that the activation phenotypes of three positive control lines (MB011B and SS40755) could be observed after olfactory conditioning sessions. We added this data as Figure 2-figure supplement 2. For SS75200 and SS77383, a split-GAL4 driver for MBON33, we observed a loss of activation phenotype in the second trial of LED ON/OFF binary choice assay (Figure 3H). Therefore, we reran the 10s LED activation experiments without a previous optogenetic olfactory conditioning assay; these data are now also included in Figure 2-figure supplement 2.

**Reviewer #2 (Recommendations For The Authors):**
Below, I list some comments and suggestions which I hope could help the authors further improve their manuscript.1. The authors identified 2 candidate lines for MBON28. It would be helpful if they could clarify how they determined whether a split-GAL4 correctly labels an MBON or is just a candidate line.

We have added in the methods section an explanation of the criteria used.

“The correspondence between the morphologies of EM skeletons and light microscopic images of GAL4 driver line expression patterns was used to assign GAL4 lines to particular cell types. This can be done with confidence when there are not multiple cell types with very similar morphology. However, in the case MBON28 we were not able to make a definitive assignment because of the similarity in the morphologies of MBON16, MBON17 and MBON28.”

2. The authors have previously shown that the expression pattern of a GAL4 driver is strongly influenced by the reporter used. The expression patterns of the split-GAL4 lines in this study are based on 20XUAS-Chrimson-mVenus trafficked (attp18), the expression strength of which may differ from other reporters or effectors. I suggest that the authors discuss this potential caveat in their manuscript. This will allow readers to be more cautious and check the expression patterns with their own reporters/effectors when using these new split-GAL4 lines.

We added the sentences below to address this concern.

“The expression patterns shown in this paper were obtained using an antibody against GFP which visualizes expression from 20xUAS-CsChrimson-mVenus in attP18. Directly visualizing the optogenetic effector is important since expression intensity, the number of labeled MBONs and off-targeted expression can differ when other UAS-reporter/effectors are used (for an example, see Figure 2—figure supplement 1 of Aso et al., 2014a).”

3. For the kinematic parameters in Fig. 2C, it is important to also show the baseline value of the parameters (i.e., the value before the light stimulation). For example, if a group of flies moves slower during the baseline period, their slower speed during the light-on period may not be due to MBON activation.

Figure 2 has been revised to include the z-scores for the 2s period just before turning on LED. The source data includes the parameter values used to calculate z-scores.

4. For Methods and Materials, the authors mostly refer to previous papers or websites for details. However, it would be helpful if they could include in this manuscript key information essential for repeating their experiments, such as the reporter/effector transgenes, empty-split controls, and antibodies and their working concentrations. It would also be helpful if they could provide the manufacturers and catalog numbers for the reagents used in this study.

We have added Appendix 1- Key Resource Table to list all the key reagents.

5. The original studies that identified the reward or punishment dopaminergic neurons mentioned in this manuscript should be cited.

We have added the following citations:

“Total number of synaptic connections from each MBON type to DANs and OANs. Based on the valence of memory when activation of DANs is used as unconditioned stimulus in olfactory conditioning (Aso et al., 2012, 2010; Aso and Rubin, 2016; Claridge-Chang et al., 2009; Huetteroth et al., 2015; Ichinose et al., 2015; Lin et al., 2014; Liu et al., 2012; Yamada et al., 2023; Yamagata et al., 2016, 2015)”